# Effects of Physical Activity and Circadian Rhythm on SCL-90 Scores by Factors among College Students

**DOI:** 10.3390/bs13070606

**Published:** 2023-07-21

**Authors:** Huimin Li, Yong Zhang

**Affiliations:** Department of Physical Education, College of Physical Education, Anhui Polytechnic University, Wuhu 241000, China; 2211310125@stu.ahpu.edu.cn

**Keywords:** physical activity, circadian rhythm, SCL-90, psychological disorder, college students

## Abstract

Objective: A study was conducted to investigate the effects of different levels of physical activity and circadian rhythm differences on the nine factors of obsessive-compulsive disorder, interpersonal sensitivity, depression, anxiety, hostility, phobia, paranoia, and psychoticism on the SCL-90 scale. Methods: A questionnaire and mathematical and statistical methods were used to conduct the study. Data were collected through a web-based cross-sectional survey of college students from three universities in Anhui. A statistical analysis of the collected data was conducted using mathematical and statistical methods. Results: A total of 1248 students were included in the statistics of this study. Binary logistic regression analysis revealed that low physical activity levels were associated with somatization (OR = 1.36, 95% CI = 0.95–1.94), obsessive-compulsive disorder (OR = 1.85, 95% CI = 1.25–2.75), interpersonal sensitivity (OR = 1.94, 95% CI = 1.30–2.88), depression (OR = 2.03, 95% CI = 1.31–3.16), anxiety (OR = 1.67, 95% CI = 1.03–2.69), hostility (OR = 1.80, 95% CI = 1.12–2.89), phobia (OR = 1.88, 95% CI = 1.20–2.94), and paranoia (OR = 2.23, 95% CI = 1.43–3.46). Circadian rhythm differences were associated with somatization (OR = 0.91, 95% CI = 0.87–0.96), obsessive-compulsive disorder (OR = 0.93, *p* < 0.01, 95% CI = 0.89–0.98), interpersonal sensitivity (OR = 0.90, 95% CI = 0.85–0.94), depression (OR = 0.92, 95% CI = 0.87–0.97), anxiety (OR = 0.89, 95% CI = 0.83–0.95), hostility (OR = 0.91, 95% CI = 0.86–0.97), phobia (OR = 0.87, 95% CI = 0.82–0.93), and paranoia (OR = 0.90, 95% CI = 0.85–0.95) were all negatively associated. In addition, gender was associated with somatization and obsessive-compulsive disorder (OR = 0.75, 95% CI = 0.57–0.98), depression (OR = 0.92, 95% CI = 0.87–0.97), and paranoia (OR = 0.55, 95% CI = 0.40–0.76). Conclusions: Low-intensity physical activity was more likely to be associated with somatization, obsessive-compulsive disorder, relationship sensitivity, depression, anxiety, hostility, terror, and paranoia than high-intensity and moderate-intensity physical activity, and circadian rhythm differences showed that people who slept later (known as nocturnal) were more likely to have these problems.

## 1. Introduction

From the data made public by the World Health Organization in 2022, we can learn that 1 billion people in the world suffer from mental disorders, accounting for about one-eighth of the global population. Of these, about 240 million people suffer from depression and 374 million from anxiety disorders [1]. Depression is ranked as the leading cause of disability worldwide [2]. Every 40 s, someone loses his or her life due to depression, which has caused a serious public health care burden. Mental health disorders are not just “mood disorders”, as they are commonly known, but are caused by a complex interaction of genetic, neuro-biomechanical, endocrine, and social factors and are characterized by prolonged depressed mood, sleep disturbances, feelings of self-ignorance, fatigue, and low energy, and in severe cases, even self-harm or suicidal thoughts.

Decreased physical activity [3] and sleep disturbances [4] are two important factors in the onset of mental health disorders, but they are not the only ones. Daily behavioral rhythms (e.g., motor activity) and daily physiological rhythms (e.g., sleep and appetite) are also often disrupted during depression [5,6]. High-energy food consumption resulting from the unhealthy eating habits of the Western lifestyle is associated with poor mental health [7]. However, consumption of foods such as fruits, vegetables, unsaturated fats, and foods rich in vitamin D was significantly and positively associated with mental health [7,8]. Poor health lifestyles such as low physical activity, poor diet, smoking, alcohol, and substance abuse are often found in patients with serious mental illness [9]. That mentally passive sedentary behavior (for example, watching TV) [10] also contributes to the increase of the symptoms. In a study of Chinese adolescents, time spent online or playing video games, as well as heavy study or homework outside of school, negatively impacted adolescents’ self-rated health and psychological well-being. Additionally, screen time has the greatest impact on mental health [11]. Although there are numerous ways to prevent and treat mental health disorders (e.g., medication, interpersonal psychotherapy, etc.) as economic and medical technology develops, 75% of patients in low- and middle-income areas still do not receive treatment [12].

Several studies have shown physical activity to be effective in reducing the incidence of depression and anxiety, and its treatment guidelines recommend exercise as a treatment strategy for managing depressive symptoms [13]. The effect of physical activity on mental health improvement has been found to be significant in various populations. In a study of Brazilian adolescents, emotional well-being was positively correlated with physical activity and negatively correlated with screen time [14]. Studies have found that physical activity can improve depressive symptoms in older adults [15]. After a minimum of four weeks of physical activity intervention, older adults showed improvements in scores on the Anxiety State, Trait Anxiety, and Anxiety Self-Assessment scales [16]. Cai et al. [17] found that physical activity during pregnancy reduced the prevalence and severity of preconception depression and anxiety. A study by Goodwind et al. shows that people without daily physical activity have a higher risk of depression than those who are physically active [18]. In addition, physical activity is considered a cost-effective intervention, both for the prevention and control of the condition [19] or as an adjunct to treatment for mental health disorders [20]. Suicide rates among patients can be reduced through exercise [21], and this is even more evident in critically ill patients. In addition, exercise was associated with a significant reduction in OCD [22]. The factors that improve mental health disorders with exercise are related to the hormones secreted by the body during exercise. Both dopamine, secreted in large amounts at the beginning of exercise [15], and endocannabinoids, secreted in large amounts during sustained exercise, can reduce anxiety and depression after exercise and make people feel relaxed and happy.

Circadian rhythms are approximately 24 h endogenous oscillations that influence and regulate the chronology of almost all human behavior and physiology [23]. A large body of research suggests that circadian rhythm disturbances may be a causal factor in some depressions (major depression, bipolar disorder, etc.) [24,25]. Eveningness is a common circadian phenotype in depression, with some studies reporting an increased risk of suicidal behaviors and more severe depressive symptoms among evening types than among morning types [26,27,28]. Early morning people have lower suicide rates than night people, which may be related to a causal relationship between early morning and improved mental health [29]. Although strong associations with depression [30], obsessive compulsive disorder (OCD) [31], and somatization [32] have been found in circadian rhythm sleep-wake disorder phase disorders, not too many studies were found on the chronological type of circadian rhythm differences with other psychological disorders other than depression.

The SCL-90 score is divided into a total score and a factor score. A total score of more than 160 or a factor score of more than two needs to be considered a positive screening, and a positive score indicates a possible mental health problem, and a higher score indicates a higher degree of mental health problem. Additionally, the nine factors of the scale represent the distribution characteristics of individual symptoms. In previous studies, most studies have focused on the effects of physical activity and sleep on somatization, obsessive-compulsive depression, anxiety, and other symptoms, and few studies have addressed interpersonal sensitivity, hostility, phobia, paranoia, and psychotic symptoms. The present study examines the effects of physical activity and circadian rhythm differences on OCD, interpersonal sensitivity, depression, anxiety, hostility, phobia, paranoia, and psychoticism through a cross-sectional study.

## 2. Materials and Methods

### 2.1. Subjects Recruitment

This study was conducted as an online cross-sectional survey among undergraduate students at three universities in Anhui Province. Because of the cross-regional nature of the study, the “Questionnaire Star” software was used to facilitate data collection. For the students to fill out the questionnaire correctly, the physical education teacher of each class explained the content of the questionnaire before the class and informed the participating students of the purpose of filling out the questionnaire, and to obtain the consent of all the subjects. To ensure the validity of the data filled in, the questionnaire was completed by the physical education teacher during the class to supervise its completion. A total of 1300 questionnaires were collected, and after excluding incomplete and invalid questionnaires, the sample size included in the analysis was 1248. Among them, 608 were male students and 640 were female students.

### 2.2. Test Indicators

The questionnaire content included the International Physical Activity Questionnaire-Short Form (IPAQ-SF), the Morning and Evening Questionnaire-5 Items (MEQ-5), and the Symptom Check List 90 (SCL-90) scales, as well as basic socio-demographic information.

#### 2.2.1. Physical Activity

Physical activity levels are judged using the International Physical Activity Questionnaire, or IPAQ, which was first introduced in Geneva in 1998. The scale is divided into an International Physical Activity Questionnaire Long Form (IPAQ-LF) with 27 questions and an International Physical Activity Questionnaire Short Form (IPAQ-SF) with only 7 questions. The questionnaire includes information about daily work, daily life, daily transportation, sports and exercise, and leisure activities. This study used self-reported data using the International Physical Activity Questionnaire Short Form (IPAQ-SF), which requires respondents to report the frequency and duration of each physical activity to the nearest hour and minute. The scale classifies physical activity levels into high-intensity physical activity levels, moderate-intensity physical activity levels, and low-intensity physical activity levels. The IPAQ-SF has been shown to have good reliability and validity in previous studies [33].

#### 2.2.2. Circadian Rhythm Differences

The circadian rhythm difference index was tested using the morningness-eveningness questionnaire (MEQ). In 2006, three Chinese researchers succeeded in introducing a Chinese version of the MEQ [34], an internationally accepted tool for measuring the natural tendencies of sleep-circadian rhythms, in a detailed version with 19 items (MEQ-19) and a simple and convenient version with 5 items (MEQ-5). In this study, the concise version of the MEQ-5 was used in the questionnaire design, and this Chinese version of the scale has been validated for reliability and validity [35]. The MEQ-5 classifies subjects according to their sleep-wake habits into early morning chronotype (early sleep and early awakening type), intermediate chronotype (normal type), and late sleep chronotype (late sleep and late awakening type), with a total score range of 4 to 25 points.

#### 2.2.3. Symptom Check List 90 (SCL-90)

The SCL-90 score is widely used to measure clinical psychiatric symptoms and mental health status and is intended for people over 16 years of age [36]. The questionnaire requires respondents to recall and assess their own psychological and physical conditions over the past week, answering 90 questions from nine dimensions. The SCL-90 is primarily unidimensional and shows sufficient item measure invariance to be a useful tool for screening adolescents for overall psychopathology [37].

### 2.3. Mathematical and Statistical Methods

This study was analyzed using SPSS Statistics version 26, using binary logistic regression analysis to explore the strength of correlations between physical activity levels and circadian rhythm differences on somatization, obsessive-compulsive disorder, interpersonal sensitivity, depression, anxiety, phobia, hostility, paranoia, and neuroplasticity.

## 3. Results

### 3.1. Somatization

As shown in Table 1, low physical activity levels were positively associated with somatization (OR = 2.04, *p* < 0.001, 95% CI = 1.36–2.75), meaning that people with low physical activity levels were 2.04 times more likely to have somatization compared to those with moderate physical activity and high physical activity; the indicator of circadian rhythm differences showed that those with eveningness were 0.91 times more likely to have somatization than those with morningness (OR = 0.91, 95% CI = 0.87–0.96); from the gender perspective, men were 1.43 times more likely to have somatization than women (OR = 1.43, *p* < 0.05, 95% CI = 1.09–1.88).

### 3.2. Obsessive-Compulsive Disorder

As seen in Table 2, low physical activity levels were positively associated with somatization (OR = 1.85, *p* < 0.01, 95% CI = 1.25–2.75), indicating that people with low physical activity levels were 1.85 times more likely to have OCD than those with moderate and high physical activity; indicators of differences in night rhythms showed that those who had eveningness were 0.93 times more likely to have OCD than those who had morningness (OR = 0.93, *p* < 0.01, 95% CI = 0.89–0.98); from the gender point of view, women were 0.75 times more likely to have OCD symptoms compared to men (OR = 0.75, *p* < 0.05, 95% CI = 0.57–0.98).

### 3.3. Interpersonal Sensitivity

As seen in Table 3, low physical activity levels were positively associated with interpersonal sensitivity (OR = 1.94, *p* < 0.01, 95% CI = 1.30–2.88), indicating that people with low physical activity levels were 1.94 times more likely to have symptoms of interpersonal sensitivity than those with moderate and high physical activity; indicators of circadian rhythm differences showed that people with a late sleep chronotype are more likely to develop interpersonal sensitivity situations. The eveningness population has a higher risk of having interpersonal sensitivity than the morningness population, which is 0.90 times higher. (OR = 0.90, *p* < 0.001, 95% CI = 0.85–0.94).

### 3.4. Depression

Table 4 shows that low physical activity levels were significantly and positively associated with depression (OR = 2.04, *p* < 0.001, 95% CI = 1.36–2.75), which means that compared to people with moderate and high-intensity physical activity levels, those with low physical activity levels had a higher risk of developing depressive symptoms by up to 2.03 times; the indicator of circadian rhythm differences showed that the late sleep chronotype was the more likely they were to suffer from depression. The risk of depressive symptoms in people with eveningness was 0.92 times higher than in people with morningness. (OR = 0.92, *p* < 0.01, 95% CI = 0.87–0.97); In terms of gender, women had a higher risk of depression than men, 0.63 times more likely (OR = 0.64, *p* < 0.01, 95% CI = 0.46–0.86). 

### 3.5. Anxiety

Table 5 shows that low physical activity levels were positively associated with anxiety (OR = 1.67, *p* < 0.05, 95% CI = 1.03–2.69), indicating that people with low physical activity levels were more likely to have symptoms of anxiety compared to those with moderate and high levels of physical activity and that low levels of physical activity were 1.67 times more likely to suffer from anxiety than moderate and high levels of physical activity; indicators of differences in circadian rhythms showed that people with eveningness were more likely to have anxiety than those with morningness. Additionally, those with a late sleep chronotype had a 0.89-fold higher risk of anxiety than those with an early morning chronotype (OR = 0.89, *p* < 0.001, 95% CI = 0.83–0.95).

### 3.6. Hostility

As seen in Table 6, low physical activity level was positively associated with hostility (OR = 1.80, *p* < 0.05, 95% CI = 1.12–2.89), indicating that people with low physical activity level were 1.8 times more likely to be hostile than those with moderate and high physical activity; the indicator of circadian rhythm differences showed that people with a late sleep chronotype were more likely to be hostile than those with an early morning chronotype. The tendency to develop hostility was 0.91 times higher in the eveningness than in morningness. (OR = 0.91, *p* < 0.01, 95% CI = 0.86–0.97).

### 3.7. Phobia

As shown in Table 7, low physical activity level was significantly and positively associated with phobia (OR = 1.88, *p* < 0.01, 95% CI = 1.20–2.94), indicating that people with low physical activity levels increased the likelihood of experiencing panic by 1.88 times compared to moderate and high levels of physical activity; the circadian rhythm difference indicator showed that people with eveningness were more likely than those with morningness to experience phobic symptoms (OR = 0.87, *p* < 0.001, 95% CI = 0.82–0.93).

### 3.8. Paranoia

Table 8 indicates that low physical activity levels were significantly and positively associated with paranoia (OR = 2.23, *p* < 0.001, 95% CI = 1.43–3.46), meaning that people with low physical activity levels had higher levels of paranoia compared to those with moderate physical activity and high physical activity by a factor of 2.23; indicators of circadian rhythm differences showed that people with eveningness were 0.90 times more likely to have paranoid symptoms than those with morningness (OR = 0.92, *p* < 0.001, 95% CI = 0.85–0.95). Women were 0.55 times more likely to suffer from paranoia than men (OR = 0.55, *p* < 0.001, 95% CI = 0.40–0.76).

## 4. Discussion

There is increasing interest in research on the relationship between exercise and mental health, but current research has focused more on depression and anxiety symptoms, with only a small number of studies on obsessive-compulsive disorder, somatization, interpersonal sensitivity, hostility, terror, and paranoia. The present article is the first study to combine physical activity and circadian rhythm differences to produce effects on somatization, obsessive-compulsive disorder, interpersonal sensitivity, depression, anxiety, hostility, terror, paranoia, and neuroplasticity, but the results of physical activity and circadian rhythm differences on psychopathic patients were not statistically significant.

### 4.1. Physical Activity

Scholars have found that for every standard deviation increase in physical activity level, symptoms of somatization disorder decrease by 10.7% [38]. The results of this study were the same as theirs. However, this article also found that men were more likely to have somatization symptoms than women. Regarding obsessive-compulsive disorder, the results of the current study showed that moderate and high-intensity physical activity possessed lower obsessive-compulsive symptoms among college students, with women having more pronounced obsessive-compulsive symptoms than men. Although no effect of differences in physical activity levels on obsessive-compulsive symptoms was found, there are arguments for the improvement of OCD with different intensities of exercise. Two studies by Abrantes et al. showed that aerobic exercise lasting at least 20 min and moderate-intensity continuous training up to at least 150 min per week significantly reduced symptoms of anxiety, depression, and obsessive-compulsive disorder [39,40]. Interpersonal sensitivity was only found in a Moroccan study of factors associated with physical activity and mental health during the closed quarantine of the COVID-19 pandemic, which showed that low-intensity recreational exercise did not bring about significant changes in interpersonal relationships, while the higher the intensity of exercise, the more pronounced the improvement in symptoms [41]. The results of the current study showed that for each standard deviation increase in physical activity, the phenomenon of interpersonal sensitivity decreased by 1.94 times. In addition, college students with low physical activity levels were more likely to experience depression and anxiety symptoms relative to moderate and high-intensity students, a result identical to that of the previous authors. Ghrouz et al. found that moderate and high-intensity physical activity levels were significantly and negatively associated with depression and anxiety [42]. In terms of gender, the results of the study by Grasdalsmoen et al. showed that men were more prone to depressive tendencies than women at low intensity levels [43]. In contrast, this paper found that women had a higher risk of developing depression than men at low-intensity activity levels. In addition, there is less research on the effects of differences in physical activity levels on hostility, phobia, and paranoia symptoms. Although some studies suggest that the idea that greater hostility is associated with less physical activity in populations with low social support is the same as the results of the current study, further research is needed on the relationship between physical activity and hostility. Aerobic exercise and physical activity showed some improvement in symptoms of phobia and paranoia, but more detail was not given on the intensity and duration of exercise. It is necessary to conduct an experimental study to explore the positive effects of physical activity on hostility, phobia, and paranoia symptoms.

### 4.2. Circadian Rhythm Differences

In a review of sleep chronotype, circadian rhythms, and mood, Bauducco et al. found that regardless of study design and measurement type, people with late sleep patterns had a higher risk of depression [44]. This view is the same as the results of the present study. Regarding the effect of circadian rhythm differences and anxiety symptoms in adolescents, the findings of related researchers are also consistent with this paper in that late sleep chronotype students are more likely to be anxious and have poorer sleep quality than early morning chronotype students [45]. In patients hospitalized with OCD, it was found that altering the sleep-wake phase delay by controlling the time of lights out significantly improved obsessive-compulsive symptoms, more so in severe patients [46]. In conducting the literature review and collection, no studies have been found on the correlation between circadian rhythm differences and symptoms of interpersonal sensitivity, hostility, fear, or paranoia. However, the results of the present study showed that for each standard deviation decrease in circadian rhythm variance score, there was a corresponding increase in interpersonal sensitivity, hostility, phobia, and paranoia.

### 4.3. Limitations

I.The data was collected through questionnaires, which came from the respondents’ subjective feelings, which may have certain biases;II.The sample size is not large enough, covering a relatively small area of colleges and universities, and the data will be continuously tracked and improved in the future to continue the study.

### 4.4. Future Research Directions

Future research directions should use controlled experimental approaches to condition interventions corresponding to physical activity and circadian rhythm differences to explore what beneficial effects would occur in people with mental health problems when both are equally effective.

## 5. Conclusions

Our study found that both physical activity levels and differences in circadian activity have an impact on multiple factors related to mental health. These factors include somatization, obsessive-compulsive disorder, relationship sensitivity, depression, anxiety, hostility, phobias, and paranoia. Low-intensity physical activity was found to be more likely to cause these symptoms than high or moderate-intensity physical activity. Individuals with evening chronotypes were also more likely to experience these symptoms. The study also found that men were more likely to experience somatization symptoms, while women were more likely to experience symptoms related to OCD, depression, phobias, and paranoia.

## Figures and Tables

**Table 1 behavsci-13-00606-t001:** Physical activity, circadian rhythm differences, and gender to somatization.

Variables	B	*p*	EXP (B)	95% CI	
Low physical activity	0.715	0.000 ***	2.04	1.36	3.05
Eveningness	−0.087	0.000 ***	0.91	0.87	0.96
Gender	0.361	0.010 **	1.43	1.09	1.88

Notice: *** At the 0.001 level (two-tailed), the correlation is significant. ** At the 0.01 level (two-tailed), the correlation is significant.

**Table 2 behavsci-13-00606-t002:** Physical activity, circadian rhythm differences, and gender to obsessive-compulsive disorder.

Variables	B	*p*	EXP (B)	95% CI	
Low physical activity	0.619	0.002 **	1.85	1.25	2.75
Eveningness	−0.068	0.006 **	0.93	0.89	0.98
Gender	−0.287	0.039 *	0.75	0.57	0.98

Notice: ** At the 0.01 level (two-tailed), the correlation is significant. * At the 0.05 level (two-tailed), the correlation is significant.

**Table 3 behavsci-13-00606-t003:** Physical activity and circadian rhythm differences to interpersonal sensitivity.

Variables	B	*p*	EXP (B)	95% CI	
Low physical activity	0.664	0.001 **	1.94	1.30	2.88
Eveningness	−0.103	0.000 ***	0.90	0.85	0.94

Notice: *** At the 0.001 level (two-tailed), the correlation is significant. ** At the 0.01 level (two-tailed), the correlation is significant.

**Table 4 behavsci-13-00606-t004:** Physical activity, circadian rhythm differences, and gender to depression.

Variables	B	*p*	EXP (B)	95% CI	
Low physical activity	0.711	0.002 **	2.03	1.31	3.16
Eveningness	−0.080	0.004 **	0.92	0.87	0.97
Gender	−0.456	0.004 **	0.63	0.46	0.86

Notice: ** At the 0.01 level (two-tailed), the correlation is significant.

**Table 5 behavsci-13-00606-t005:** Physical activity and circadian rhythm differences to anxiety.

Variables	B	*p*	EXP (B)	95% CI	
Low physical activity	0.515	0.034 *	1.67	1.03	2.69
Eveningness	−0.113	0.000 ***	0.89	0.83	0.95

Notice: *** At the 0.001 level (two-tailed), the correlation is significant. * At the 0.05 level (two-tailed), the correlation is significant.

**Table 6 behavsci-13-00606-t006:** Physical activity and circadian rhythm differences to hostility.

Variables	B	*p*	EXP (B)	95% CI	
Low physical activity	0.591	0.014 *	1.80	1.12	2.89
Eveningness	−0.088	0.004 **	0.91	0.86	0.97

Notice: ** At the 0.01 level (two-tailed), the correlation is significant. * At the 0.05 level (two-tailed), the correlation is significant.

**Table 7 behavsci-13-00606-t007:** Physical activity and circadian rhythm differences to phobia.

Variables	B	*p*	EXP (B)	95% CI	
Low physical activity	0.631	0.006 **	1.88	1.20	2.94
Eveningness	−0.129	0.000 ***	0.87	0.82	0.93

Notice: *** At the 0.001 level (two-tailed), the correlation is significant. ** At the 0.01 level (two-tailed), the correlation is significant.

**Table 8 behavsci-13-00606-t008:** Physical activity, circadian rhythm differences, and gender to paranoia.

Variables	B	*p*	EXP (B)	95% CI	
Low physical activity	0.803	0.000 ***	2.23	1.43	3.46
Eveningness	−0.099	0.000 ***	0.90	0.85	0.95
Gender	−0.587	0.000 ***	0.55	0.40	0.76

Notice: *** At the 0.001 level (two-tailed), the correlation is significant.

## Data Availability

The data presented in this study are available on request from the corresponding author. The data are not publicly available due to subject privacy contained in the data.

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
