# Peer review of "Effects of Physical Activity and Circadian Rhythm on SCL-90 Scores by Factors among College Students"

_behavsci, 2023, doi:10.3390/bs13070606_

Round 1

Reviewer 1 Report

This article attempts to assess an important aspect of contemporary medicine. It is known that the circadian rhythm has age-specific characteristics and, as is well known, social requirements usually act contrary to the chronotypic preferences of adolescents and students. Moreover, modern trends in the development of physical inactivity are alarming, as it can also play a significant role in the development of various diseases. However, in its current form, the manuscript has major comments that need to be revised before being published.

lines 65-74: The flow of this paragraph is generally not logical. 

The authors state that "There is now a large body of research that suggests that circadian rhythm disturbances may be a causal factor in some depression" - What types of depression did you mean? Or maybe depression syndrom or depression-like behaviour?

Authors state that "Delayed sleep-wake phase disorder and depression, Obsessive Compulsive Disorder (OCD), Somatization a strong correlation was also confirmed. In addition, delayed sleep-wake phase disorders can lead to cardio-vascular, cancer, inflammatory diseases, and cognitive impairment." - The authors discuss the importance of sleep disorder, which is a separate disease and not a feature of the circadian rhythm. 

Authors use "circadian rhythm differences" in the manuscript, but it must be changed on the chronotype or circadian type, because it is a major mistake.

Authors state that "late sleep is more likely to trigger depression" - Not a late night sleep, but an eveningness as a characteristic of chronotype. This is a fundamental error that is also found throughout the text.

 lines 75-77:

"The cumulative sum of the total scores of the Symptom Check List90 can tell the severity of the patient's depressive condition," - only 1 subscale of SCL-90 describe depression severity.

lines 77-80:

Need the references for this: "In previous studies, most studies 77 have focused on the effects of physical activity and sleep on somatization, obsessive-compulsive depression, anxiety, and symptoms, and few studies have addressed interpersonal sensitivity, hostility, Phobia, paranoia, and psychotic symptoms."

lines 109-112: 

Authors have to use write title of the scale -  morningness-eveningness questionnaire - and add scoring for this. Why did the authors use two chronotype scales?

It is not clear what the authors used in the regression analysis called "Circadian differences". 

line 268:

Don`t use "time type" because Authors in this references researched Chronotype.

lines 286-294 should be improved:

for example, 

Our study found that both physical activity levels and differences in circadian activity have an impact on multiple factors related to mental health. These factors include somatization, obsessive-compulsive disorder, relationship sensitivity, depression, anxiety, hostility, phobia, and paranoia. The low-intensity physical activity was found to be more likely to cause these symptoms than high or moderate-intensity physical activity. Additionally, individuals with (evening chronotypes? later chronotypes?) were also more likely to experience these symptoms. The study also found that men were more likely to experience somatization symptoms, while women were more likely to experience symptoms related to OCD, depression, phobia, and paranoia.

The manuscript needs ongoing revision for errors and flow.

For example:

lines 66-68

"There is now a large body of research that suggests that circadian rhythm disturbances may be a causal factor in some depression." may be improved as "Recent research indicates that disruptions to our circadian rhythm may contribute to the development of depression."

lines 68-69:

"Delayed sleep-wake phase disorder and depression 68 [18], Obsessive Compulsive Disorder (OCD) [19], Somatization [20] a strong correlation 69 was also confirmed. " may be improved as 'It has been confirmed that there is a strong correlation between delayed sleep-wake phase disorder and depression, Obsessive Compulsive Disorder (OCD), and Somatization."

lines 77:

In previous studies, most studies - shoud be rephrase

lines 130-131:

moderate physical activity and high physical activity; indicators of circadian rhythm differences showed that people who slept later (night The indicator of circadian rhythm differences showed that those who slept later (nighttime population) 

The whole manuscript need major revision of quality of writing.

Author Response

Dear Reviewer:

Thank you for reading my article carefully and attentively and giving every suggestion in detail, your suggestions have been very beneficial to me. I have also annotated my article in the revision, and I hope it will facilitate you to read my article again.
For the suggestions you gave:

  1. What type of depression is meant by some depression? I have added in my article (major depression, bipolar disorder, etc)
  2. Regarding your point that delayed sleep-wake phase disorder is not a feature of circadian rhythms, I have made a distinction in the article.
  3. Regarding your point that "circadian rhythm differences" are used in the manuscript but must be modified in terms of temporal type or circadian rhythm type, I did not make a distinction between temporal types of circadian rhythm differences in the article at the beginning, so I did not express it well enough. Significantly I have modified all of them in the revised version.
  4. The phrase "people who sleep late are more likely to be depressed" was also misunderstood because of incorrect expression. What I actually wanted to express was that eveningness is more likely to cause depressive symptoms, and this has been revised.
  5. The meaning of the total scl-90 score in lines 75-77 has also been revised.
  6. Revised the reference you gave for lines 77-80.
  7. Your suggestion for 109-112 was due to the fact that at the beginning I did not express clearly in the methodology the scales used, extremely scoring and analysis criteria. The addition has been made in the latest version.
  8. The wording error in line 268 has been corrected and changed to "chronotype".
  9. Took your comments on lines 286-294 for improvement and corrected them.
  10. Removed parts 130-131 that you suggested deleting.

Finally, thank you again for reading this article with care and making changes, and I wish you a happy life.

Sincerely,
Huimin Li

Reviewer 2 Report

Dear authors,

I enjoyed reading your paper. Simple and precise.

I'll only have 2 suggestions:

1st - I'll add, in lines 45 to 47, that 'mentally passive sedentary behavior (for example, watching TV)' * (Santos, et al., 2022) also contributes to the increase of the symptoms. And then, after line 51, you can again use the same paper to link the benefits of physical exercise in reducing the incidence of depression.

Santos, J., Ihle, A., Peralta, M., Domingos, C., Gouveia, É. R., Ferrari, G., ... & Marques, A. (2022). Associations of physical activity and television viewing with depressive symptoms of the European adults. Frontiers in public health9, 799870.

2nd - Your limitation is quite intriguing. I don't think it is a limitation but a 'future research' opportunity.  

Good luck in your future. 

Author Response

Dear reviewer:

Thank you for reading my article carefully and carefully and giving me beneficial suggestions. Thank you very much for expressing your liking for my article, which has given me a lot of confidence in my research path.
For the suggestions you gave:

  1. Add Santos' relevant literature on mental passive sedentary behavior in lines 45-47, which I have added in the revised version.
  2. In line 51 I have also added the relevant literature to demonstrate the beneficial effects of physical activity on mental health.
  3.  I have also made a distinction between limitations and future research directions in the revised version.

Finally, thank you again for reading my article carefully and giving careful suggestions. I wish you a happy life!

Sincerely,
Huimin Li

Round 2

Reviewer 1 Report

The authors have made the necessary corrections to the manuscript, so now it can be recommended for publication.